🔓 | **Open Peer Review** | Bacteriology | Research Article

# Genomic and metabolic insights into the first host-associated isolate of *Psychrilyobacter*

**Meijia Liu,**[1] **Guangshan Wei,**[1,2] **Qiliang Lai,**[1] **Zhaobin Huang,**[1] **Min Li,**[3] **Zongze Shao**[1,2]

**ABSTRACT** Although gut bacteria are vital to their hosts, few studies have focused on marine animals. *Psychrilyobacter* is frequently related to various marine animals, but its interaction with host remains unknown due to the lack of host-associated isolate or genomic information. Here, we combined cultivation-independent and cultivation-dependent methods to uncover the potential roles of *Psychrilyobacter* in the host abalone. The high-throughput sequencing and literature compiling results indicated that *Psychrilyobacter* is widely distributed in marine and terrestrial ecosystems with both host-associated and free-living lifestyles, but with a strong niche preference in the guts of marine invertebrates, especially abalone. By *in vitro* enrichment that mimicked the gut inner environment, the first host-related pure culture of *Psychrilyobacter* was isolated from the abalone intestine. Phylogenetic, physiological, and biochemical characterizations suggested that it represents a novel species named *Psychrilyobacter haliotis* B1. Carbohydrate utilization experiments and genomic evidence indicated that B1 can utilize diverse host-food-related monosaccharides and disaccharides but not polysaccharides, implying its potential role in the downstream fermentation instead of the upstream food degradation in the gut. Particularly, this strain showed potential to colonize the gut and benefit the host via different strategies, such as the short-chain fatty acids generation by fermenting peptides and/or amino acids, and the putative production of diverse vitamins and antibiotics to support the host growth and antipathogenicity. To our knowledge, strain B1 represents the first host-related pure culture of *Psychrilyobacter*; genomic and metabolic evidence showed some beneficial characteristics of the dominant gut anaerobe to the host.

**IMPORTANCE** *Psychrilyobacter* is a globally distributed bacterial genus and with an inhabiting preference for guts of marine invertebrates. Due to the difficulty of cultivation and the limited genomic information, its role in host remains largely unknown. We isolated the first host-associated *Psychrilyobacter* species from abalone gut and uncovered its functional potential to the host through different mechanisms. Our findings provide some insights into the understanding of host-microbe interactions on a core taxon with the marine invertebrates, and the isolate may have an application potential in the protection of marine animals.

**KEYWORDS** *Psychrilyobacter*, gut bacteria, pure culture, microbe-host interaction

The impacts of gut microbes on host growth and health have been extensively studied, especially the key microbial genera of humans, which have been confirmed to interact with the host through various mechanisms (1, 2). However, studies on the relationship between marine animals and their gut microbes are very limited. Abalone, as a typical representative of marine shellfish, is widely distributed in coastal regions and is also raised as a marine food product all over the world. Abalone mainly feeds on marine macroalgae (red and brown algae) that are digested in the intestine and digestive gland

Address correspondence to Zongze Shao, shaozz@163.com.

Meijia Liu and Guangshan Wei contributed equally to this article. The names' order was decided by the surname alphabetically.

The authors declare no conflict of interest.

See the funding table on p. 16.

(3, 4). The abalone gut usually presents a microaerobic or anaerobic internal environment (5), assuming that anaerobic microorganisms are involved in the digestion and health of the host (6–8).

Previous culture-independent studies have observed the composition of the abalone intestinal microbiome (9–11). The genus *Psychrilyobacter* is a key member with high relative abundance of over 20% in the intestines of European abalone (*Haliotis tuberculata*) regardless of the season and feeding diet (3). Recently, a high-throughput sequencing study has also shown that the proportion of *Psychrilyobacter* accounted for approximately 15–30% of all bacteria in the gut of healthy *Haliotis diversicolor* (11); similar to *Haliotis discus hannai*, this genus is more enriched in faster-growing large abalone (10). Interestingly, *Psychrilyobacter* is also widely distributed in various other marine animals such as oysters (12, 13), sea vases (14), Atlantic salmon (15), deep-sea snails (16), green-lipped mussels (17), Chilean mussels (18), and even deep-sea hydrothermal vent crabs (19). At the same time, *Psychrilyobacter* members are also distributed in non-host habitats, such as anoxic seawater and marine sediments, but are usually members of the "rare biosphere" with extremely low abundance (20, 21). Therefore, *Psychrilyobacter* members possess a close association with diverse marine animals, but their role in the host remains unexplored.

Previous studies with environmental enrichments indicated that *Psychrilyobacter* is involved in the primary hydrolysis and fermentation of *Spirulina*, as one of the most prominent protein degraders in the subarctic marine sediment (22, 23). To date, only a few environmental pure cultures of this genus have been reported, with origins of marine environments exclusively. For example, a marine sediment-derived isolate featured obligately anaerobic utilization of a broad range of complex organic substrates (21). Similarly, strains isolated from the anoxic Black Sea water columns support that *Psychrilyobacter* members are important detrital organic matter degraders in marine environments (20, 21). Because of the absence of host-associated pure culture and corresponding genome information, the interaction of *Psychrilyobacter* with hosts and its roles in marine animals remain unknown.

This study aims to enrich and isolate host-related *Psychrilyobacter* and explore its roles in the host gut. Considering that *Psychrilyobacter* is particularly abundant in the abalone intestine, *H. discus hannai* was chosen for bacterial enrichment and isolation. Pure culture of the first host-related *Psychrilyobacter* will provide both genomic and metabolic evidence to decipher its relationship with the host. This study will help to promote the understanding of the ecological functions of dominant taxa in the marine animal intestinal microbiome and provide culturable resources for the development of marine animal intestinal probiotics.

## RESULTS AND DISCUSSION

### *Psychrilyobacter* dominated the abalone gut and was widely distributed in diverse marine animals

To assess the abundance of *Psychrilyobacter*, the bacterial community compositions in intestinal samples of *H. discus hannai* and *H. diversicolor* were detected using high-throughput sequencing of the near full-length 16S rRNA gene. At the phylum level, *Fusobacteria* was one of the most dominant taxa in all samples, with relative abundances of 10.3–39.4% among different individuals (Fig. 1A). The dominant bacterial genera in the *H. discus hannai* intestine were mainly affiliated with *Psychrilyobacter* occupying relative abundances of 32.9–38.0%, followed by *Vibrio* (23.2–47.8%) and *Mycoplasma* (10.9–33.6%) (Fig. 1B). Similarly, the bacteria dominating the intestinal microbiome of *H. diversicolor* were mainly *Psychrilyobacter* (10.3–39.0%) and *Mycoplasma* (6.7–41.9%) (Fig. 1B). Meanwhile, previous studies on gastrointestinal bacterial communities from abalones with different species (3, 9–11, 24), geographical locations (3, 9, 10, 24, 25), and diets (3, 11, 24) also support that *Psychrilyobacter* is a core predominant member, even reaching over 90% in some species, such as *H. tuberculate* (3).

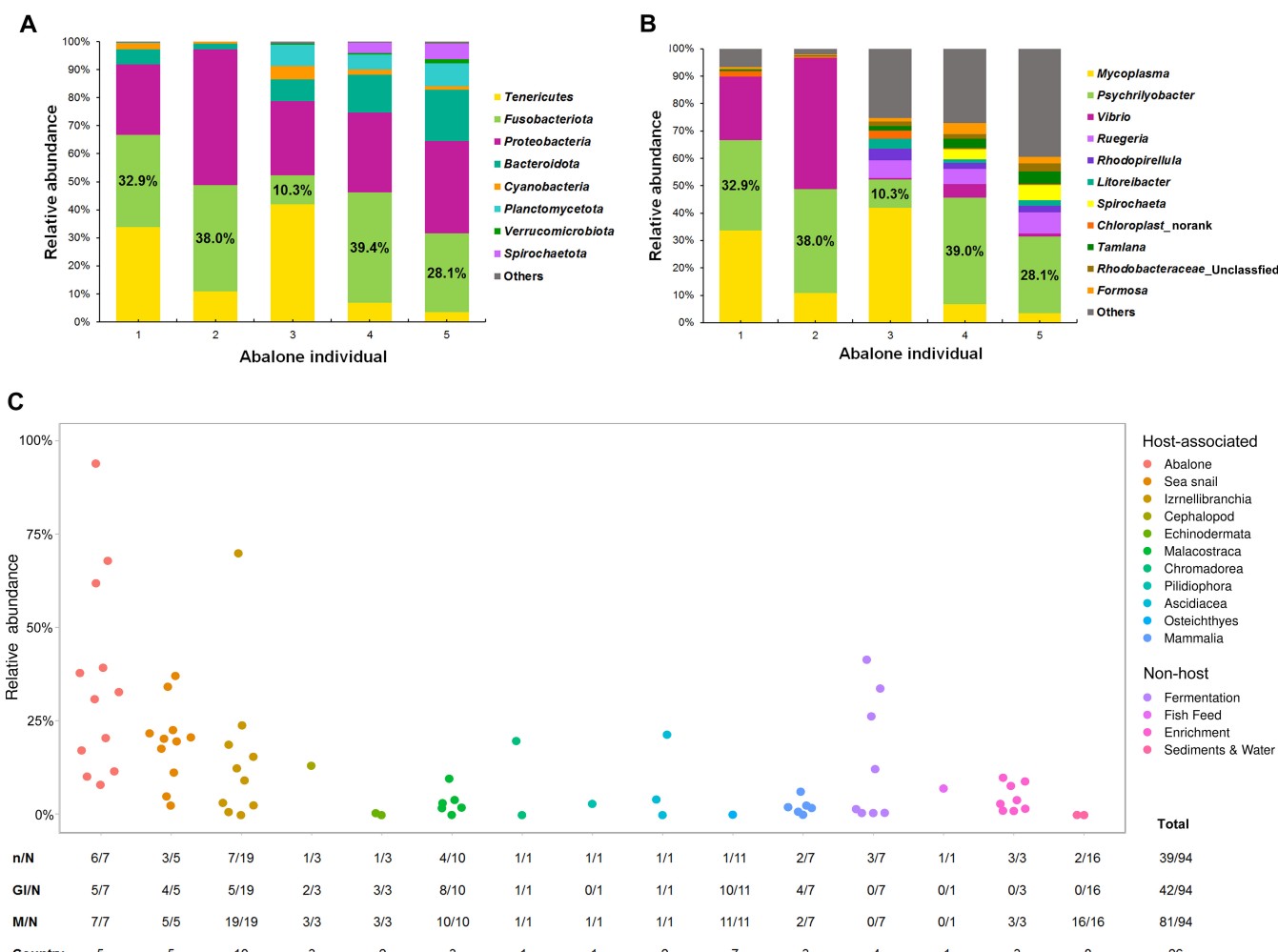

FIG 1  Bacterial community compositions in the sampled abalones' guts and the distribution of *Psychrilyobacter* in different hosts and environments. (A) The relative abundance of gut bacteria in the sampled abalone at the phylum level. (B) The relative abundance of gut bacteria in the sampled abalone at the genus level. The abalone individual numbers 1 and 2 are *H. discus hannai*, and the individual numbers 3–5 are *H. diversicolor*. The values of relative abundance of the Fusobacteriota phylum and the *Psychrilyobacter* genus are shown in panels A and B, respectively. (C) The compiled distribution pattern of *Psychrilyobacter* across different habitats all over the world. The relative abundances present all data that we can collect from the published papers including the mean, minimum, or maximum values of *Psychrilyobacter*. "n" represents the number of papers that have the exact relative abundances of *Psychrilyobacter*; "N" represents the number of papers that mentioned *Psychrilyobacter*; "GI" is the number of studies that sampled from gastrointestinal tracts; and "M" represents the number of studies that sampled from marine habitats. This dotplot was created by PlotsOfData (https://huygens.science.uva.nl/PlotsOfData/).

To further clarify the distribution pattern of *Psychrilyobacter*, nearly 100 published articles about "*Psychrilyobacter*" (1 May 2022) were compiled, summarizing the relative abundances, hosts, and sample locations (Fig. 1C; Table S1). These results showed that *Psychrilyobacter* is widely distributed in terrestrial and marine ecosystems with both host-associated and free-living lifestyles (Fig. 1C). Generally, members of *Psychrilyobacter* are inclined to be associated with marine invertebrates and are most abundant in abalone. With respect to inhabiting niches in marine animals, the gastrointestinal tract is preferred first, followed by the digestive gland and gill (Table S1). In addition, in a few artificial environments, such as fermented foods ( smelly mandarin fish and cheese) and protein-rich enrichments, *Psychrilyobacter* was also detected as one of the dominant members (Table S1). Marine sediments and water columns also host *Psychrilyobacter* but merely as a rare species (Fig. 1C; Table S1). Therefore, *Psychrilyobacter* is a widespread genus and is associated with various marine animals with a preference for invertebrates, especially the

high abundance in the intestine of global abalones, implying underlying coevolution with the host.

## Enrichment and pure culture of *Psychrilyobacter*

To obtain pure cultures of *Psychrilyobacter*, bacteria were enriched through simulating the *in situ* intestinal environment of *H. discus hannai*, including anaerobic, lower temperature (15°C), weakly acidic pH (6, 7), and various host food-related polysaccharides (algae, agar powder, and alginate) or yeast extract plus tryptone as the substrates, respectively. After 2 weeks of enrichment, the 16S rRNA gene clone library was constructed to identify the dominant taxa in different enrichments. The obtained results indicated that *Vibrio* was dominant in all four enrichments, and *Psychrilyobacter* was abundant in three enrichments, exhibiting the highest proportion (40%) in medium with the presence of yeast extract plus tryptone (Fig. 2A; Table S2).

Bacterial isolation was further performed in an anaerobic chamber on a marine agar plate under 15°C. In total, 63 pure bacterial cultures within 16 genera were obtained from the enrichments with four different substrates, of which most isolates (44 strains of 14 genera) were from the enrichment with yeast extract plus tryptone (Fig. 2B and C). The taxonomical compositions of these isolates were significantly different from those of the enriched communities. Except for the members of *Vibrio* that were isolated in all enrichments, many culturable bacteria were not detected in the enrichments (Fig. 2A and B). As a result, only two *Psychrilyobacter* isolates, designated strain sp. B1 and sp. W3, were obtained from the enrichments with yeast extract plus tryptone (Fig. 2C; Table S3). Our results suggest that the host abalone food-related polysaccharides did not perform well in the enrichment and isolation of *Psychrilyobacter*, which is further supported by

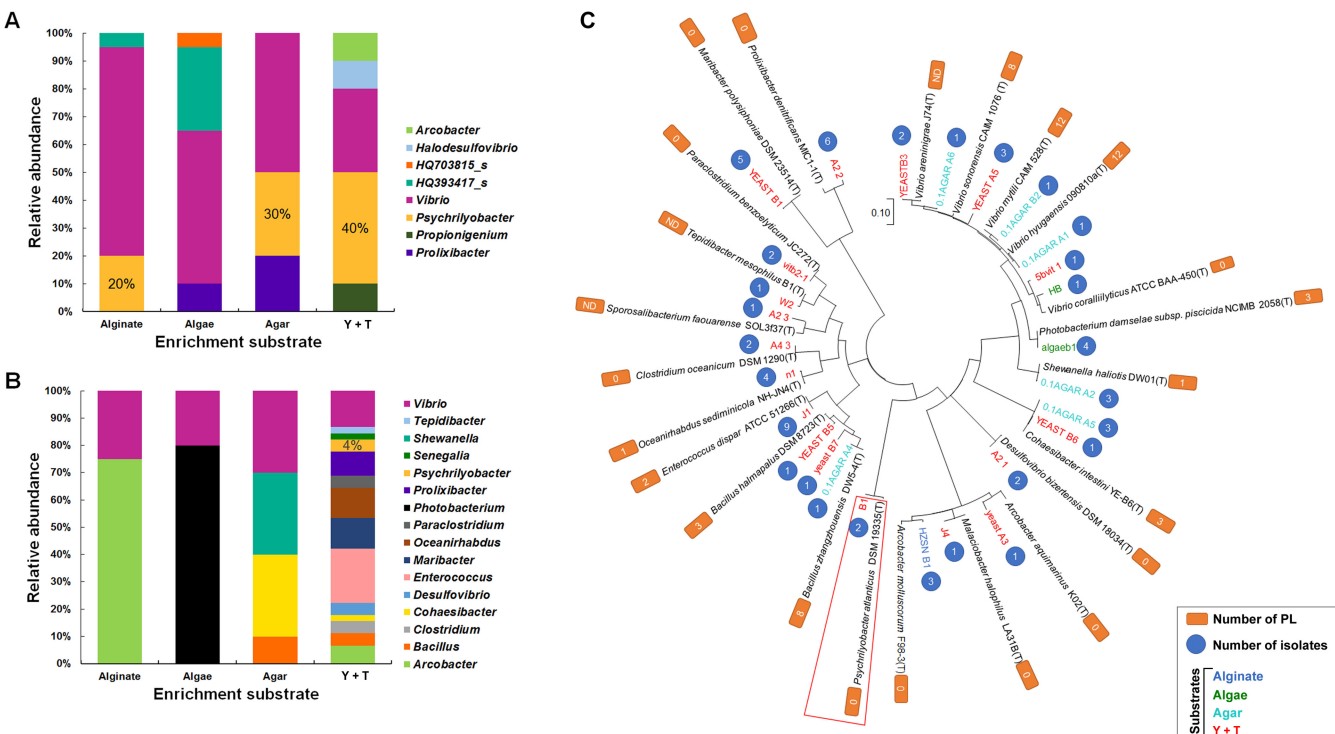

**FIG 2** Bacterial compositions of enrichments and isolations using different substrates. Panels A and B show the genus-level bacterial compositions in the enrichments and isolations, respectively. (C) Species-level phylogenetic tree of all the isolates based on their 16S rRNA genes. The isolates with more than 99% similarities were combined, and one of them was randomly selected for the tree construction. The *Psychrilyobacter* isolates and their closest type strain are marked with a red box. The closest type strains were obtained through similarity searches against the EzBioCloud Database. Genomes of the type strains were downloaded from the NCBI genome database and were used to annotate the polysaccharide lyase (PL) in the CAZy database. The "Y + T" means the substrate of "yeast extract plus tryptone."

the polysaccharide lyase (PL) counts in the genomes of most related type strains of the pure cultures from distinct enrichment substrates (Fig. 2C). Meanwhile, these results imply that *Psychrilyobacter* is still hard to cultivate on plates even though it shows high abundances in guts and enrichments. Moreover, its growth may be facilitated by other bacteria or hosts, although the non-host derived relatives can be detected in marine environments, such as in subarctic sediment, in which anaerobic bacterial degradation of protein and lipid macromolecules occurs (23), but simply as a rare species. Further alignment indicated that the two *Psychrilyobacter* strains were identical in the 16S rRNA gene sequence, and strain B1 was chosen for the following analyses.

## Phylogenetic identification of the first host-related *Psychrilyobacter*

After sequence similarity search in the EzBioCloud Database, the 16S rRNA gene sequence of strain B1 has maximum sequence similarity with *Psychrilyobacter atlanticus* DSM19335[T] (99.6%) isolated from marine sediment of the Atlantic Ocean, followed by *Psychrilyobacter piezotolerans* SD5[T] (98.6%) from sulfidic seawater of the Black Sea. We also compared the 16S rRNA gene of strain B1 with sequences >1,400 bp in length and over 97% similarity from the NCBI Nucleotide collection database (26). The obtained results showed that the relatives of *Psychrilyobacter* were widely distributed in the ocean with both free-living and host-associated lifestyles (Table S4). Of note, there are only five reported isolates in the genus *Psychrilyobacter*, including *P. atlanticus* DSM19335[T] from marine sediment (21), *P. piezotolerans* strain SD5[T] and BL5 from seawater (20), and *Psychrilyobacter* sp. strain STAB 703 and STAB 704 from marine biofilms of feldspar and quartz minerals (27). To our knowledge, strain B1 represents the first host-associated pure culture of *Psychrilyobacter*.

Further phylogenetic analyses based on both the 16S rRNA genes and genomes of the type strains of *Fusobacteriaceae* were performed to determine their exact taxonomic status (Fig. 3). Strain B1 in the phylogenetic tree of 16S rRNA genes was placed within the genus *Psychrilyobacter* containing all type strains found thus far with minimal differences in the lengths of branches, which is separated from other clusters within the genera *Ilyobacter*, *Propionigenium*, *Fusobacterium*, and *Cetobacterium* (Fig. 3A). Furthermore, the genomic tree showed that strain B1 forms a monophyletic branch distinctly separated from both type strains of the genus *Psychrilyobacter* (Fig. 3B). Therefore, strain B1 represents a potential novel species affiliated to the genus *Psychrilyobacter* within the *Fusobacteriaceae* family of the Fusobacteria phylum.

Importantly, the computation results of strain B1 and the two type strains showed that the ANI values ranged from 85.6% to 88.0% and the dDDH values ranged from 42.2% to 63.6%, which are far below the standard criteria of ANI (95.0–96.0%) and dDDH (70%) for delineation of novel bacterial species (28, 29). These values strongly supporting that strain B1 is a novel species of *Psychrilyobacter*, for which the name *Psychrilyobacter haliotis* (ha.li.o'tis. N.L. gen. n. *haliotis* named after the scientific name of the abalone *Haliotis*) sp. nov. is proposed. The new type strain is strain B1[T] (=MCCC 1A14957).

Intriguingly, relatives of strain B1 in the trees showed host-associated or free-living lifestyles in marine or terrestrial environments, and the novel species *P. halioits* represents the uniquely host-associated pure culture in the marine-originating *Psychrilyobacter* members (Fig. 3A and B).

## Phenotypic characteristics of the novel bacterium *P. haliotis* B1

Strain B1 can anaerobically grow on the Marine Agar 2216 plate (BD Difco) and forms translucent colonies with a diameter of 0.5–2 mm, smooth and moist surface, and neat edge after cultivation at 15℃ for 24 h. The cells of strain B1 are Gram-negative. TEM observation revealed an ellipsoid cell with a length of 1.2–2.8 μm and a width of 0.8–1.2 μm and without flagellum (Fig. 4). Compared with the two type strains, the cell size of B1 is larger than strain HAW-EB21[T] but much shorter than strain SD5[T] (Table 1). Strain B1 can grow at 4–28℃ with the optimum of 18℃, pH of 5–11 with the optimum of 6–7, and salinity at 1–4% with the optimum of 3% (Table 1). Generally, the optimal growth

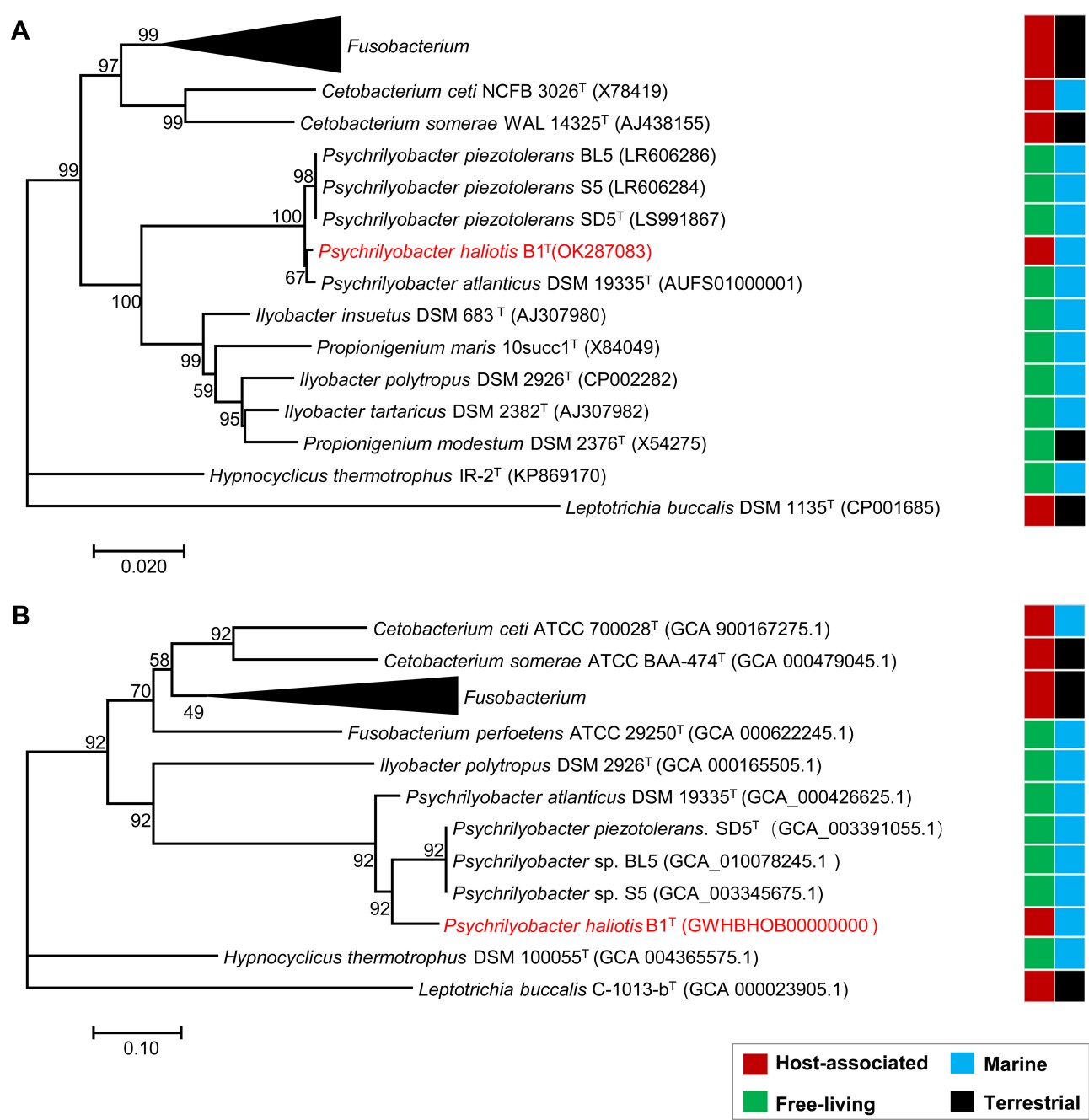

**FIG 3** The 16S rRNA gene-based (A) and genome-based (B) phylogenetic trees of strain B1. In the 16S rRNA gene-based tree, the sequences of 20 type strains of *Fusobacterium* were compressed to the black triangle. The 14 type-strain genomes of *Fusobacterium* were compressed to the black triangle in the genome-based phylogenetic tree. Our research target strain B1 is marked in red. In the bar graph behind the phylogenetic trees, the first column represents the lifestyle of the strains, and the second column represents the sources of the strains.

temperature of B1 is similar to the sea sediment isolate HAW-EB21[T] with low-temperature preference and lower than the deep-sea sulfidic water isolate SD5[T]. Their similar optimal salinities implied that all strains exhibit marine or saline habitat preferences. Distinctively, strain B1 showed higher tolerance to pH changes and preferred acidic conditions compared with the other two non-host-derived type strains, which may fit the relatively harsh environments from alkaline external to acidic abalone digestive tracts (e.g., with pH 5.3–6.6 for *Haliotis laevigata*) (5). In addition, the profiles of the fatty acid composition of strain B1 are also very different from those of the two type strains, with higher fatty

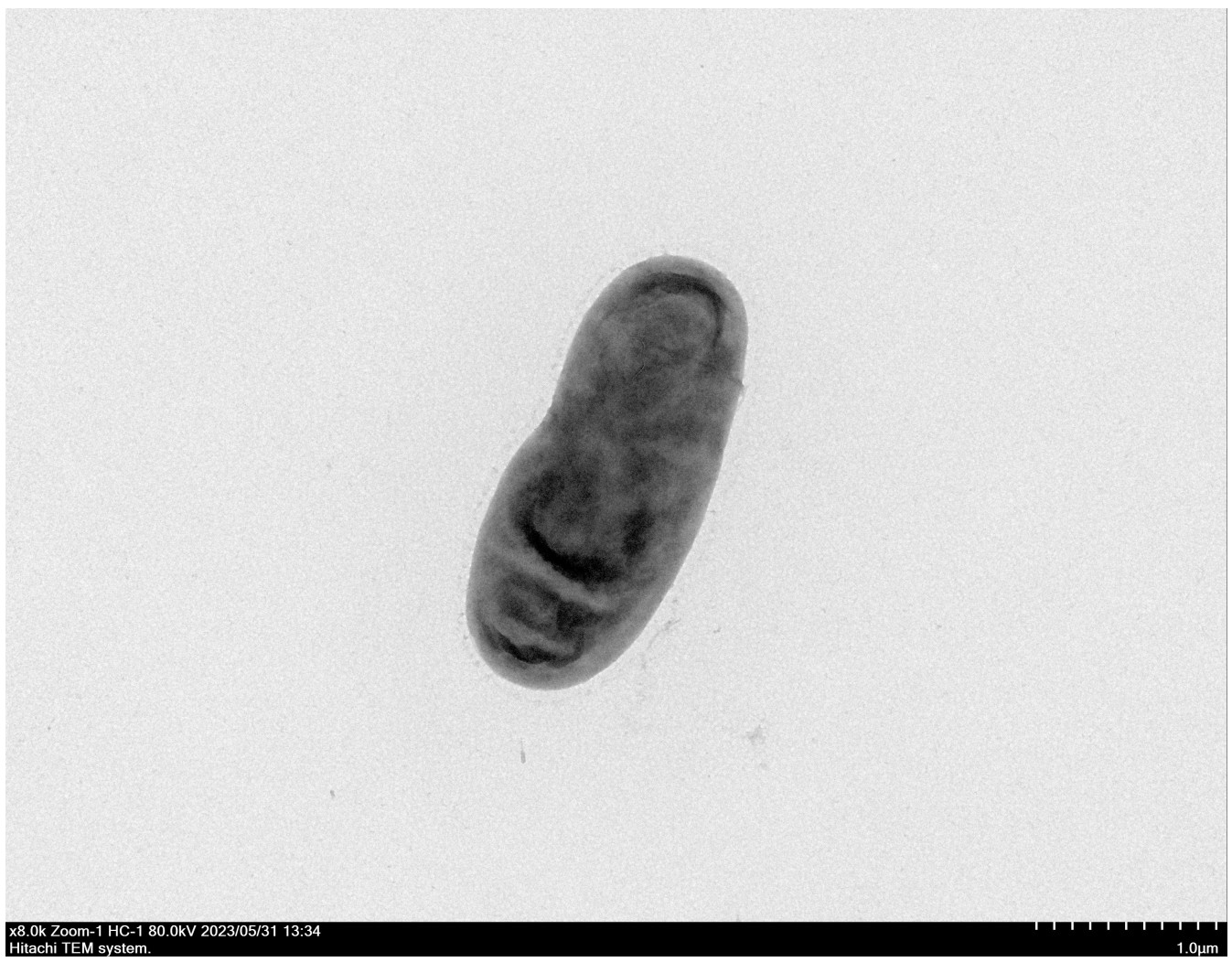

```
x8.0k Zoom-1 HC-1 80.0kV 2023/05/31 13:34
Hitachi TEM system.                                                      1.0µm
```

**FIG 4**  Cell morphology of strain B1 grown under optimal growth condition. The cell morphology was observed by transmission electron microscopy (TEM) after negative staining. The machine parameters and the scale bar are shown at the bottom.

acids of $C_{11:0}$, $C_{13:0}$, $C_{15:1}$ $\omega 6c$, and $C_{17:1}$ $\omega 8c$ and lower fatty acids of $C_{12:0}$ 3OH, $C_{18:1}$ $\omega 11c$, and $C_{18:0}$ 3OH (Table S5). Collectively, all the unique phenotypic characteristics reconfirmed that strain B1 represents a novel species within the genus *Psychrilyobacter* and suggested its distinctive preference for the host-associated lifestyle.

### Genomic features of *P. haliotis* B1 implied complex lifestyles

To further explore the genomic features and potential relationships with its host, the complete genome of strain B1 was assembled. Interestingly, as the first complete genome of *Psychrilyobacter*, strain B1 has three circular replicons with different sizes. According to a previous definition (30), the three replicons can be classified into a chromosome, a putative chromid, and a plasmid (Fig. 5). Briefly, the chromosome is the largest and contains most of the core genes; the chromid features the largest secondary replicon, with similar GC content to the chromosome and carrying some core genes, while the plasmid has the smallest size, the lowest GC content and very few core genes (31, 32) (Fig. 5; Table S6).

The chromid encodes genes of the chromosome partitioning system (*parA* and *parB*) and many core essential genes, such as central carbohydrate metabolism and various amino acid metabolism, and includes one housekeeping gene, *pyrG* (Fig. 5; Table S6),

**TABLE 1** Physiological and biochemical properties of strain B1 in compared with the closely related strains

| Characteristics | 1[a] | 2[b] | 3[c] |
|---|---|---|---|
| Cell size length × width (µm)[d] | 0.8–1.2 × 1.2–2.8 | 0.5 × 0.5–1 | 0.6–0.7 × 3.0–6.0 |
| Flagellum | No | No | No |
| Temperature range (°C)[d] | 4–28 (18) | 4–27.5 (18.5) | 4–35 (20–25) |
| pH range for growth[d] | 5–11 (6–7) | 5.5–9.5 (7) | 6.5–8.8 (7–8) |
| NaCl range (%, wt/vol)[d] | 1–4 (3) | 0.5–2 (2) | 0–5.5 (2–3) |
| Isolating source | Abalone intestine | Marine sediment | Sulfidic seawater |

[a]Strain B1.
[b]*P. atlanticus* HAW-EB21[T].
[c]*P. piezotolerans* SD5[T].
[d]Phenotypic characteristics were obtained from Zhao et al. (21) (*P. atlanticus* HAW-EB21[T]), and Yadav et al. (20) (*P. piezotolerans* SD5[T]). The values in the bracket represent the optimum conditions.

which consists of the definition of chromid (30, 34). Importantly, previous studies have shown that bacteria with chromids have very distinct lifestyles, and most are associated with animal or plant hosts (30). Meanwhile, evidence from transcriptomes has suggested that chromid-borne genes are critical to the shift between free-living and host-associated states (35, 36). Moreover, the chromid also has two rRNA genes (Fig. 5), which may lead to chromosome-chromid fusion through multiple rRNA operons in bacterial genomes to adapt to harsh conditions (37). Therefore, it is supposed that strain B1 gains both free-living and host-associated lifestyles and shifts between the internal and external environments of abalones, probably via conveniently switching sets of chromid genes on or off.

Compared with the other two type strains (Table S7), strain B1 shows the smallest genome size and fewer CDSs, which may be related to the coevolution with the abalone host, while the features of GC content and higher number of rRNAs and tRNAs are not coincident with the previous results on host-associated strains (20), suggesting that B1 may have both free-living and host-associated lifestyles. Strain B1 has 30 copies of ribosomal RNA genes, which is more than the other two strains DSM19335[T] (22 copies) and SD5[T] (12 copies). The rrn copy number is highly related to the environment; a low rrn copy number is favored in oligotrophic environments, but a high rrn copy number is favored in eutrophic environments (38) and strain B1 is more adaptive to isolated sources of nutrient-rich abalone intestine. Previous deep-sea isolated members of *Psychrilyobacter* also showed potential to exhibit both free-living and host-associated lifestyles (20). Considering the distribution pattern of *Psychrilyobacter*, which generally inhabits non-host marine environments with very low abundance but host-related marine invertebrates with high abundance (Fig. 1; Table S1), as well as the abovementioned genomic features, this genus is proposed to be characterized with double lifestyles and prefer host-associated state when they encounter marine hosts.

## Multiple strategies for interactions with the abalone host

To further understand the relationship between *Psychrilyobacter* and the abalone host, more details related to colonization, carbohydrate metabolism, vitamin production, and defense were analyzed on the basis of the genome.

### Genes related to colonization in the abalone intestine

As one of the most abundant bacteria in the gut, the first and most important thing is colonization therein. The representative isolate strain B1, as a Gram-negative bacterium, encodes genes related to the synthesis of the lipopolysaccharide core and O-antigens at the outer membrane (Table S8), which are critical to host colonization as a bacterial gut symbiont (39). Genes of capsular polysaccharide synthesis (*rfbA*, *rfbB*, *rfbC*, and *rfbD*) are also found in the genome, which are involved in forming a protective capsule that could encourage a symbiotic relationship in the mouse gut (40). In addition, genes that enhance extracellular polysaccharide formation, such as the L-ascorbate-specific enzyme

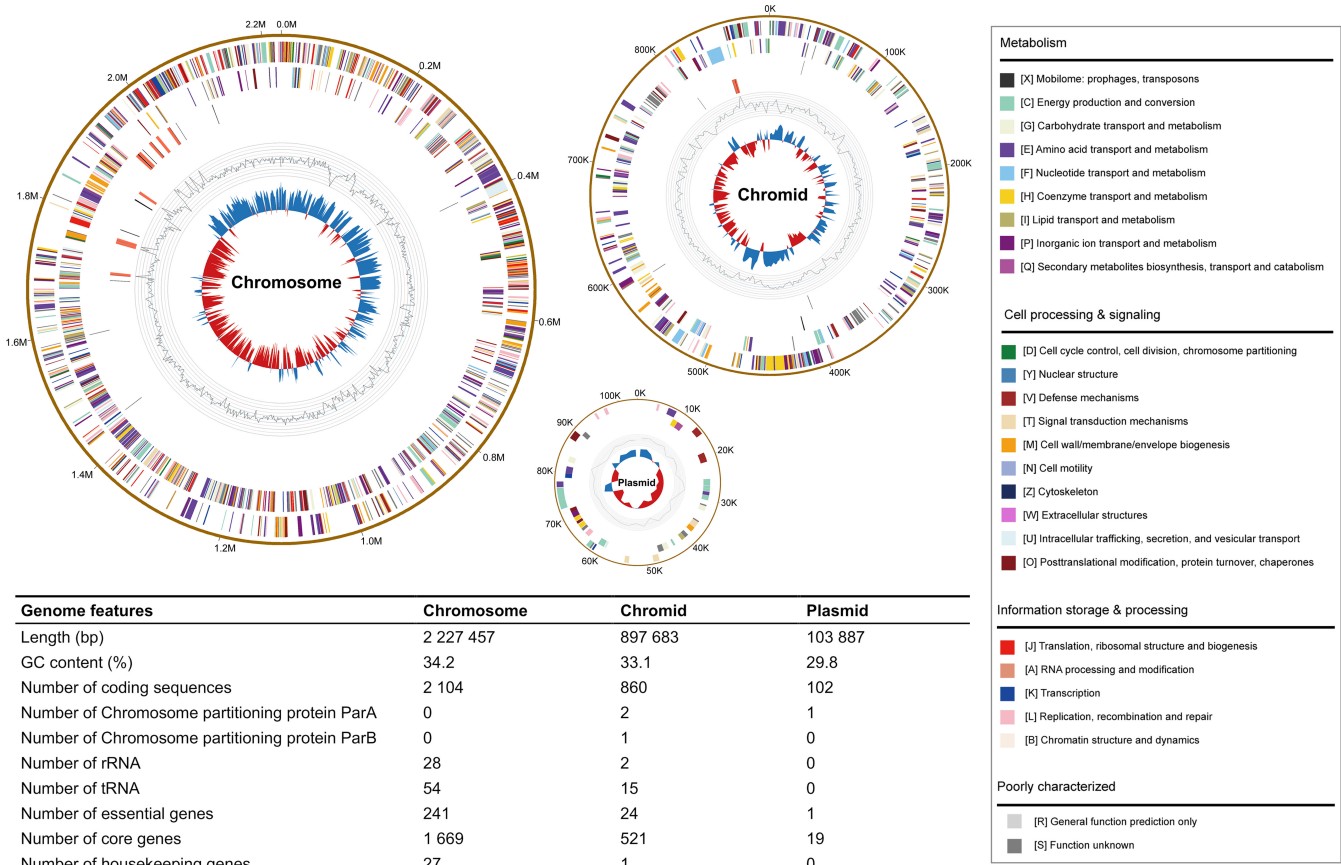

| Genome features | Chromosome | Chromid | Plasmid |
|---|---|---|---|
| Length (bp) | 2 227 457 | 897 683 | 103 887 |
| GC content (%) | 34.2 | 33.1 | 29.8 |
| Number of coding sequences | 2 104 | 860 | 102 |
| Number of Chromosome partitioning protein ParA | 0 | 2 | 1 |
| Number of Chromosome partitioning protein ParB | 0 | 1 | 0 |
| Number of rRNA | 28 | 2 | 0 |
| Number of tRNA | 54 | 15 | 0 |
| Number of essential genes | 241 | 24 | 1 |
| Number of core genes | 1 669 | 521 | 19 |
| Number of housekeeping genes | 27 | 1 | 0 |

**FIG 5** Genomic features of the three replicons of strain B1. Rings from the outermost to the center: (1) scale marks of the genome, (2) protein-coding genes on the forward strand, (3) protein-coding genes on the reverse strand, (4) tRNA (black) and rRNA (red) genes on the forward strand, (5) tRNA (black) and rRNA (red) genes on the reverse strand, (6) GC content, and (7) GC skew. Chromosome protein-coding genes are color-coded according to their COG categories. The circular plots are drawn using Circos (33) based on the genomic features.

II complex of the phosphotransferase system (PTS; *ula*A, *ula*B, and *ula*C), are encoded by strain B1 (Table S8), possibly improving bacterial tolerance against host digestion and gut acid (5, 41).

Adhesion and biofilm formation are also important for host colonization (42). Strain B1 has genes (*pilB*, *pilC*, *pilD*, *pilM*, and *pilN*) encoding type IV pilus, which can facilitate adhesion, biofilm formation, motility, and secretion (43–45). Distinctively, only the host-associated strain encodes genes for filamentous hemagglutinin (COG3210) functioning as a bacterial adhesin, which may facilitate bacterial adherence to mucosa (46). Moreover, 55 genes for quorum sensing and 27 genes for biofilm formation are encoded in strain B1, and the high abundance of these genes may secure its competitiveness during host colonization (Table S8). With respect to gene location, most of these colonization-related genes were in the chromosome (60 genes), followed by the chromid (15 genes), and only 2 of them were in the plasmid (Fig. 6B).

As the most basic aspect of microbial colonization, the formation of intestinal biofilms can enhance the synergy between bacteria and the host and improve nutrient exchange by increasing the residence time of bacteria, as well as preventing colonization of conditional pathogens by competing for attachment sites (47). Consistently, a previous study on *Ciona* supports that *Psychrilyobacter*, as a core taxon with steadily high abundance regardless of starvation status, potentially formed stable biofilms within the mucus layer (14), and the colonization of *Psychrilyobacter* in the gut of marine invertebrates requires further study to validate in the future.

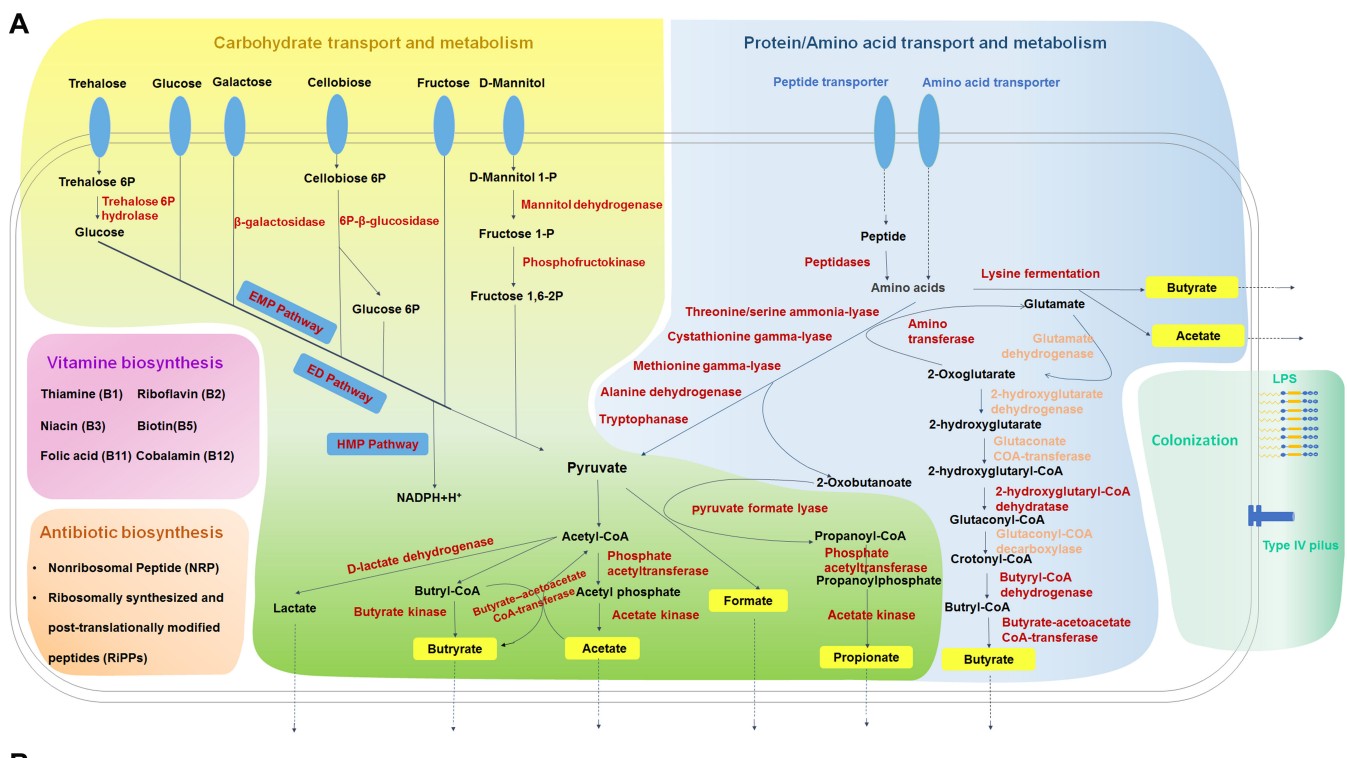

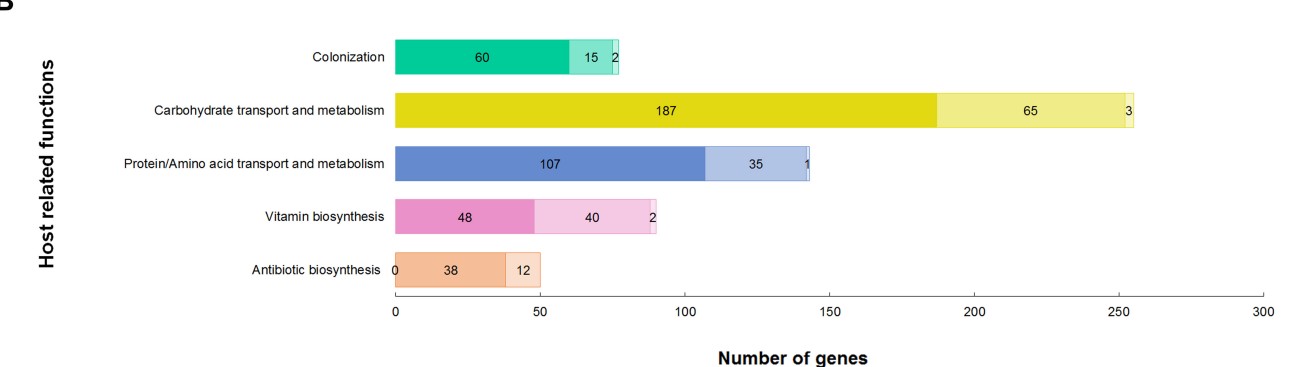

FIG 6   Putative host-related metabolic processes (A) and their distributions on the replicons (B) of strain B1. (A) From left to right, the five different blocks represent multiple mechanisms by which strain B1 may benefit the host. The two small blocks on the lower left are vitamin biosynthesis (violet) and antibiotic biosynthesis (red). The two large blocks in the middle are carbohydrate metabolism pathways (yellow) and protein/amino acid metabolism pathways (blue), and their overlapping region is the common pathways of the two parts (green). The small block on the lower right is the mechanism of intestinal colonization by strain B1 (bluish green). Black font in the blocks represents substrates or products, red font represents the enzymes corresponding to the reaction, light red font represents the unannotated enzymes, and various SCFAs are highlighted in the blocks. The solid arrow represents the reaction direction, and the dotted arrow represents outward diffusion. The blue ellipsoids in the membrane represent related transporters. More details on the pathways are listed in Tables S8, 13 through16. (B) The bar chart represents the number of genes occupied by the three replicons of each metabolic block in Fig. 6A. From left to right (the color from deep to light), each part represents chromosome, chromid, and plasmid, respectively.

## Carbohydrate metabolism and short-chain fatty acid production

Abalone mainly feeds on different macroalgae. To determine the activities of carbohydrate degradation, diverse carbohydrate-active enzymes in the strain B1 genome were identified according to the CAZy database. A total of 115 genes involved in carbohydrate metabolism were retrieved from the genome, which is higher than those from the other two type strains of *Psychrilyobacter* (Table S9). These carbohydrate-active enzymes (CAZymes) include 2 auxiliary activities, 9 carbohydrate esterases, 10 carbohydrate-binding modules, 41 glycoside hydrolases, 52 glycosyl transferases, and 1 PL. In contrast,

no PL is encoded in the genomes of the other two type strains of *Psychrilyobacter*, *P. atlanticus* HAW-EB21$^T$, and *P. piezotolerans* SD5$^T$ (Table S9).

KEGG and COG annotations predicted that strain B1 could utilize various monosaccharides and disaccharides such as D-ribose, D-xylose, D-glucose, D-fructose, D-galactose, D-mannose, D-mannitol, maltose, lactose, trehalose, and sucrose (Tables S10 and S11). A variety of carbohydrates related to the composition of algae (48, 49) were selected to verify the metabolic ability (Table S12). As a result, B1 could use all selected carbohydrates as the sole carbon and energy source for growth and exhibited a preference of glucose > galactose > trehalose > xylose > fructose > mannitol, which confirmed its powerful metabolic capabilities of diverse monosaccharides and disaccharides. Of note, no clear growth of strain B1 was detected in algal polysaccharides as the sole carbon and energy sources, such as alginate, agar, and agarose. In the genome of B1, only one PL12_3 polysaccharide lyase was found (Table S9). BLAST results indicated that the closest homolog (58.1% similarity) was annotated as a heparinase II/III-like protein which was proven to be a component of oligoalginate lyase but could not depolymerize alginate alone (50). This indicates that B1 has the potential to assist oligoalginate degradation. The algae polysaccharides can be degraded by the enzyme produced by the host and/or other coexisting gut bacteria. For example, abalone itself can secrete HdEG66 cellulase to digest cellohexaose and cellotetraose to cellotriose, cellobiose, and glucose (51). In addition, *Vibrio halioticoli*, as one of the most dominant species detected in our abalone *in situ* gut samples, together with other members like some closely relative type strains of our polysaccharides-enriched isolates existing multiple PLs (Fig. 2C), have been proven to participate in algae polysaccharide hydrolysis directly (52, 53). As previously defined, strain B1 should play a "scavenger" role in the abalone gut, which does not produce enzymes for polysaccharide hydrolysis directly but takes up the hydrolytic products shared by other organisms (54). Therefore, *Psychrilyobacter* may not be involved in the upstream hydrolysis of host foods (algae polysaccharides) but more important in the downstream fermentation of monosaccharides and disaccharides after transformation by the host and/or other coexisting gut bacteria. This also explained the poor performance of *Psychrilyobacter* in algae polysaccharide enrichment and further isolation on agar plates (Fig. 2).

Further metabolic pathway analysis indicated that B1 could ferment diverse carbohydrates to short-chain fatty acids (SCFAs). Various carbohydrates are first converted to glucose or glucose-6-phosphate, and then are funneled into the EMP and ED pathways to produce pyruvate (Fig. 6A). Then, the pyruvate may be further transformed into diverse SCFAs and/or lactate (Fig. 6A; Table S13). Of note, the ED pathway can quickly turn glucose into two molecules of pyruvate through only four steps of metabolic reactions. Meanwhile, B1 has diverse carbohydrate-specific PTS and ATP-binding cassette transporters to facilitate carbohydrate absorption (Table S13). The above-mentioned genomic features may benefit to the competitive edge of B1 in utilization of small molecular carbohydrates in the host digestive tract. Interestingly, almost all carbohydrate metabolism genes were distributed in the chromosome and chromid, and only three of them were in the plasmid (Fig. 6B; Table S6). Moreover, strain B1 also has the potential to transform proteins, peptides, and amino acids (Table S14) into SCFAs, and most of the genes are located on the chromosome and chromid (Fig. 6A and B).

To test if the above pathways are active, high-performance liquid chromatography (HPLC) experiments were used to determine the type and concentration of organic acids fermented by strain B1 cultured with carbohydrates or tryptone. As a result, only cultured with tryptone can strain B1 to produce SCFAs, including butyric acid (323.67 ± 36.24 mg/L), propionic acid (244.78 ± 2.74 mg/L), acetic acid (244.27 ± 6.88 mg/L), formic acid (41.09 ± 2.88 mg/L), and lactic acid (12.36 ± 2.35 mg/L). While cultured with glucose, mannitol, or trehalose, the strain grew slowly and could not produce any SCFAs; only malic acid and/or tartaric acid were detected. These results indicate that strain B1 is very adept at fermenting protein and its derivatives to SCFAs, which coincides with a

previous study showing that *Psychrilyobacter* was likely involved in the primary hydrolysis and fermentation of protein-dominated *Spirulina* necromass (22).

SCFAs are essential for the growth and health of the host. For instance, a previous study showed that 1% of acetic acid plus 1% of formic acid could promote the growth of South African abalone *H. midae* (55). Acetate is speculated to provide a major energy source and may be used as a precursor for protein, sugar, and lipid synthesis in abalone (56). Acetate can also regulate glucose homeostasis and mediate gut bacteria-brain axis function in zebrafish (57). Feeding butyrate benefits growth performance and makes aquaculture animals more resistant to stressful conditions, such as juvenile grass carp (58) and juvenile *Arapaima gigas* (59). Moreover, the production of various SCFAs can help the host gut maintain acidic pH to inhibit the proliferation of opportunistic pathogens, such as pathogenic *Vibrios*, which invade the host through the gut but have low tolerance to acid and prefer alkaline conditions (60, 61). The abovementioned evidence indicates that the abalone host may benefit from the SCFAs derived from the powerful fermentation capacity of *Psychrilyobacter*.

## *Production of various vitamins essential to the host*

Vitamins are essential micronutrients that are normally found as precursors of various enzymes that are necessary for vital biochemical reactions in all living cells (62). Abalone is incapable of synthesizing most vitamins and has to be obtained exogenously. Therefore, vitamin-producing microorganisms, especially the dominant ones inhabiting the gut, may play a critical role in the health maintenance of abalone. Strain B1 has biosynthetic pathways of diverse vitamins, including thiamine, riboflavin, NAD and NADP, biotin, folate, and cobalamin, which are essential to host growth. In fish and shrimp, thiamine deficiency may result in various growth deficiency symptoms (63). Biotin acts as a coenzyme for four carboxylases during the metabolism of carbohydrates, lipids, and proteins in animals (64). In a certain range, folic acid has a positive correlation with the growth of abalone (65). Cobalamin can only be produced by certain bacterial species, and neither plants nor animals have acquired the ability to produce the vitamin themselves in the course of evolution, without the help of bacteria (66). Therefore, the powerful potential of *Psychrilyobacter* in the production of diverse vitamins may greatly benefit the growth and health of its abalone host. Intriguingly, approximately half of the genes related to vitamin biosynthesis are located on the chromid (Fig. 6B; Table S15).

## *Potential in regulating the defense and microbiome of the host*

Based on the Antibiotics and Secondary Metabolite Analysis Shell (antiSMASH) database, there are two secondary metabolite regions in the genome of B1 (Fig. S1; Table S16). One is on the chromid, which contains a putative cluster of nonribosomal peptide synthetase-like fragments and shows the highest similarity score with the isonitrile lipopeptide biosynthetic gene cluster of *Mycobacterium tuberculosis* H37Rv. The other is on the plasmid, which contains potential ribosomally synthesized and posttranslationally modified peptides and has the most similarity with bacteriocin sublancin 168 produced by *Bacillus subtilis* strain 168. Sublancin 168 exhibits bactericidal activity against Gram-positive bacteria, including important pathogens such as *Bacillus cereus*, *Streptococcus pyogenes*, and *Staphylococcus aureus* (67, 68). Similar to other invertebrates, abalone lacks an adaptive immune system and mainly depends on innate immunity (69). Thus, through the production of antibiotics, *Psychrilyobacter* showed the potential to help the host protect against the invasion of foreign bacteria and maintain intestinal microbiome homeostasis. In addition, we also found two genes encoding lysozymes on the chromid (Tables S10 and S11), which may inhibit the growth of pathogens. Interestingly, all related genes were detected in the chromid and plasmid, suggesting an irreplaceable role of the two replicons in assisting the defense of the abalone host.

Of note, although our genomic and metabolic analyses suggested various beneficial potential to the host, it cannot finally define the strain B1 as a probiotic. Because even the same species, like *Escherichia coli*, can be a pathogenic, probiotic, or commensal

strain (70). However, some previous studies also suggest that *Psychrilyobacter* species tend to be probiotic to the host. For example, Choi et al. confirmed that *Psychrilyobacter*, as one of the dominant genera, was more prevalent in large Pacific abalone (*H. discus hannai*) with higher growth rates than in small ones (10). In another study, *Psychrilyobacter* showed an overabundance in mussel individuals collected from natural conditions compared with the individuals collected from an intensive commercial culture farm (18). However, extensive researches are needed to uncover the interactions between *Psychrilyobacter* members and their diverse hosts, and future efforts will seek to obtain more isolates from different hosts and to integrate multiomics with experimental verification.

## Conclusions

In this study, the first host-associated pure culture of *Psychrilyobacter* was successfully obtained and identified as a novel bacterial species, named *P. haliotis* B1[T]. The genome of B1 was characterized as three circular replicons with one putative chromid, which provides the possibility of both free-living and host-associated lifestyles. Multiple mechanisms uncovered the potential interaction between B1 and the host, which will help us to understand the function of *Psychrilyobacter* in marine animals. Strain B1 may play a role as a "scavenger" in the gut microbiome, collaborating with other "sharing" bacteria to help the host to digest the algae foods. The bacterium is a versatile "fermenter" with powerful ability to ferment proteins, peptides and amino acids. It is also a potential key "producer" of SCFAs, vitamins, and antibiotics, which are crucial to the growth and health of the host. Although genomic evidence indicates potential interactions with the host, further investigations are needed to verify the relationship, such asfluorescence in situ hybridization (FISH), metabolomics, and germ-free animal-based experiments.

## MATERIALS AND METHODS

### Abalone sampling and intestine processing

Abalones of *H. discus hannai* and *H. diversicolor* were collected from Fuda abalone farm located in Jinjiang (118°24′E, 24°54′N), Fujian Province, China. They were put on ice and transported to the laboratory immediately. The abalones are mainly fed on brown seaweed (*Laminaria japonica*) and red algae (*Gracilaria lemaneiformis*). The shell sizes of the abalones are approximately $5.5 \pm 0.5$ cm. In the laboratory, the shell was first removed using a sterile scalpel, and then the intestine was carefully dissected with scissors. After adding 900 µL of sterile artificial seawater, the intestine of each abalone individual was thoroughly grinded to homogenate. For the bacterial diversity analysis, the individual homogenate of *H. discus hannai* and *H. diversicolor* was preserved at $-80°C$ until the DNA extraction. For the bacterial enrichment, the homogenate of two mixed individuals of *H. discus hannai* was directly used without freezing, as described below. All the abovementioned procedures were performed under sterile conditions.

### Determination of intestinal bacterial diversity of abalone

To determine the primary bacterial diversity and relative abundance of *Psychrilyobacter*, high-throughput sequencing of the nearly full-length 16S rRNA gene was performed at Shanghai Biozeron Technology Co., Ltd. (Shanghai, China) according to a previous description with a few modifications (71). The details on library construction, sequencing, and data processing are in the supplementary files.

### Intestinal bacterial enrichment and key member isolation

To determine the best medium of enrichment, four different substrates were used, including algae (the feed of abalone), agar powder (1 g/L, Solarbio, Beijing, China),

alginate (1 g/L, HUSHI, Shanghai, China), and yeast extract (1 g/L, OXOID, Hampshire, England) plus tryptone (1 g/L, OXOID, Hampshire, England) as carbon sources and energy sources, resazurin (1 g/L, Solarbio) as an oxygen indicator, and L-cysteine hydrochloride (0.5 g/L, Macklin, Shanghai, China) as a reducing agent in the artificial seawater medium under anaerobic condition (72). The intestine suspension of homogenate was inoculated into a 50-mL anaerobic flask containing a 10-mL enrichment medium. Of note, because *H. discus hannai* is a cold-water species (optimal growth temperature of 10–20°C), the culture temperature of 15°C was selected for enrichment and isolation. After 2 weeks of cultivation, the diversity of enrichments was evaluated through a 16S rRNA gene clone library method as previously described (73).

Meanwhile, 100 μL of enrichment culture of each substrate was spread on the Marine Agar 2216 plates (BD, Difco), and the culture plates were maintained at 15°C for 5 days under anaerobic condition until many visible colonies were observed. Each colony was picked and streaked onto MA plates for further purification under anaerobic condition. The 16S rRNA genes of the pure isolates were amplified and sequenced for preliminary classification against the EzBioCloud Database (74).

## Phylogenetical identification of *Psychrilyobacter*

Among the pure isolates, a strain named B1 was identified as a member of *Psychrilyobacter* by the 16S rRNA gene using the bacterial universal primers 27F and 1492R (75). For phylogenetic identification, close relatives of strain B1 were retrieved from the EzBioCloud Database (74) and NCBI nucleotide database (https://blast.ncbi.nlm.nih.gov/Blast.cgi). Sequences were aligned using CustalW, and a neighbor-joining phylogenetic tree was constructed using MEGA 7.0 (76). The node support of the tree topology was evaluated using bootstrapping estimation of 1,000 replicates.

## Complete genome sequencing and genomic relatedness comparison

Genomic DNA of strain B1 was extracted from the cells after 3 days of growth in the Marine Broth 2216 (BD Difco) using a bacterial genomic extraction kit (SaiBaisheng, Shanghai, China) according to the manufacturer's recommendation. The quality was checked using 1.0% agarose electrophoresis. The whole genome sequence of strain B1 was determined using PacBio's Single Molecule Real-Time sequencing technology (Tianjin Biochip Corporation, Tianjin, China) according to the manufacturer's instructions. The Hierarchical Genome Assembly Process (HGAP version 4) Analysis Application was used for genome assembly and data statistics (77).

The other two related genome sequences of *P. atlanticus* HAW-EB21[T] (accession number: AUFS00000000.1) and *P. piezotolerans* SD5[T] (accession number: QUAJ00000000.1) were obtained from the genome portal in the NCBI database. Digital DNA–DNA hybridization (dDDH) and average nucleotide identity (ANI) are recognized as the gold standards to define a novel bacterial species (78). The dDDH estimate value between two strains was analyzed using the genome-to-genome distance calculator (GGDC2.0) with the alignment method of BLAST (79, 80). The ANI value estimates between strain B1 and the two closely related type strains within *Psychrilyobacter* were calculated using the ANI Calculator Online Service in the EzBiocloud database (http://www.ezbiocloud.net/tools/ani) (81). Finally, a genome-based phylogenetic tree was constructed using an up-to-date bacterial core gene set consisting of 92 genes (82).

## Genome annotation and metabolic activity prediction

Gene prediction was performed using the Prodigal (Prokaryotic Dynamic Programming Gene-finding Algorithm) program (83). The rRNA and tRNA genes were identified using the barrnap (Basic Rapid Ribosomal RNA Predictor) and tRNAscan, respectively (84). The functions of putative genes were annotated using the Clusters of Orthologous Groups of proteins (85), Kyoto Encyclopedia of Genes and Genomes (86) and Rapid Annotations using Subsystems Technology (87). Carbohydrate metabolism ability was analyzed using

the CAZyme Database (88). The annotation of secondary metabolites was performed using antiSMASH (89).

The genome of strain B1 and two previously published draft genomes of *P. atlanticus* HAW-EB21[T] isolated from cold deep marine sediment and *P. piezotolerans* SD5 [T] from deep sulfidic waters were also compared through the Majorbio Cloud Platform (https://cloud.majorbio.com/) with default parameters to explore their functional differences.

## Cell morphology observation and physiological and biochemical identification

The cell morphology of strain B1 was observed by transmission electron microscopy (TEM) after negative staining. The growth conditions of strain B1, including temperature, pH, and NaCl tolerance ranges, as well as fatty acid compositions and carbohydrate utilization tests were examined, and all the details are in the supplementary files.

## Analysis of organic acids in strain B1 cultures

Four different substrates were used to cultivate strain B1, respectively, including tryptone, glucose, mannitol, and trehalose. After that, HPLC experiments were performed to test the organic acids. Some standards, like formic acid, acetic acid, propionic acid, butyric acid, and lactic acid, were utilized to identify and determine their concentrations in the samples. For each group, three biological replicates were arranged in control with the artificial seawater medium without any organic substrate. More details were listed in the supplementary files.

## ACKNOWLEDGMENTS

This work was financially supported by the project of National Natural Science Foundation of China (grant numbers 41676149 and 42030412), and the Scientific Research Foundation of Third Institute of Oceanography, MNR (grant number 2019021) to Z.S.

All authors contributed to the study conception and design. Material preparation, data collection, and analysis were performed by M.L., G.W., Z.H., and Z.S. The first draft of the manuscript was written by G.W., M.L., and Q.L., and all authors commented on previous versions of the manuscript. All authors read and approved the final manuscript.

The authors declare that they have no known competing financial interests or personal relationships that could have appeared to influence the work reported in this paper.

## AUTHOR AFFILIATIONS

[1]Key Laboratory of Marine Biogenetic Resources, Third Institute of Oceanography, Ministry of Natural Resources of the PR China; State Key Laboratory Breeding Base of Marine Genetic Resources; Fujian Key Laboratory of Marine Genetic Resources, Xiamen, China
[2]Southern Marine Science and Engineering Guangdong Laboratory (Zhuhai), School of Marine Sciences, Sun Yat-Sen University, Zhuhai, China
[3]College of Ocean and Earth Sciences, Xiamen University, Xiamen, China

## AUTHOR ORCIDs

Meijia Liu ⓘ http://orcid.org/0000-0002-4025-6508
Guangshan Wei ⓘ http://orcid.org/0000-0003-1599-3829
Zongze Shao ⓘ http://orcid.org/0000-0002-4784-090X

## FUNDING

| Funder | Grant(s) | Author(s) |
|---|---|---|
| MOST \| National Natural Science Foundation of China (NSFC) | No.41676149 | Zongze Shao |
| MOST \| National Natural Science Foundation of China (NSFC) | No.42030412 | Zongze Shao |
| Scientific Research Foundation of Third Institute of Oceanography, MNR | 2019021 | Zongze Shao |

## AUTHOR CONTRIBUTIONS

Meijia Liu, Conceptualization, Data curation, Investigation, Methodology, Visualization, Writing – original draft | Guangshan Wei, Investigation, Writing – original draft | Qiliang Lai, Investigation, Writing – review and editing | Zhaobin Huang, Conceptualization, Methodology, Project administration, Writing – review and editing | Min Li, Writing – review and editing | Zongze Shao, Conceptualization, Funding acquisition, Writing – review and editing

## DATA AVAILABILITY

Strain B1 has been deposited in the Marine Culture Collection of China (MCCC, https://mccc.org.cn/) under the accession number MCCC 1A14957. The 16S rRNA gene sequence of B1 has been deposited at DDBJ/ENA/GenBank under the accession number OK287083. The whole-genome sequence of B1 and the raw data of 16S rRNA gene high-throughput sequencing have been deposited in the Genome Warehouse in the National Genomics Data Center (https://ngdc.cncb.ac.cn/gwh/) under the accession numbers GWHBHOB00000000 and CRA006367, respectively.

## ETHICS APPROVAL

Informed consent was obtained from all individual participants included in the study.

## ADDITIONAL FILES

The following material is available online.

### Supplemental Material

**Supplemental text (Spectrum03990-22-s0001.docx).** Supplemental materials and methods.
**Supplemental tables (Spectrum03990-22-s0002.xlsx).** Tables S1 to S5, Tables S7 to S12, and Table S16.
**Table S6 (Spectrum03990-22-s0003.xlsx).** Genomic composition of strain B1.
**Table S13 (Spectrum03990-22-s0004.xlsx).** Carbohydrate metabolism-related genes in strain B1.
**Table S14 (Spectrum03990-22-s0005.xlsx).** Protein and amino acid metabolism-related genes in strain B1.
**Table S15 (Spectrum03990-22-s0006.xlsx).** Vitamin biosynthesis-related genes in strain B1.

### Open Peer Review

**PEER REVIEW HISTORY (review-history.pdf).** An accounting of the reviewer comments and feedback.

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
