## [Reviewer comments · Microbiology Spectrum]

Microbiology Spectrum

Genomic and metabolic insights into the first host-associated isolate of *Psychrilyobacter*

Meijia Liu, Guangshan Wei, Qiliang Lai, Zhaobin Huang, Min Li, and Zongze Shao

Corresponding Author(s): Zongze Shao, Third Institute of Oceanography Ministry of Natural Resources

Review Timeline:

Submission Date:	October 4, 2022
Editorial Decision:	March 25, 2023
Revision Received:	July 27, 2023
Accepted:	August 11, 2023

Editor: Konstantinos Kormas

Reviewer(s): Disclosure of reviewer identity is with reference to reviewer comments included in decision letter(s). The following individuals involved in review of your submission have agreed to reveal their identity: Muhamad Amin (Reviewer #2); Ruixuan Wang (Reviewer #3)

Transaction Report:

DOI: <https://doi.org/10.1128/spectrum.03990-22>

March 25, 2023

Dr. Zongze Shao
The Third Institute of Oceanography, State of Oceanic Administration
Key Laboratory of Marine Biogenetic Resources
Daxue Road 178#
Xiamen, Fujian 361005
China

Re: Spectrum03990-22 (The first host-associated anaerobic isolate of *Psychrilyobacter* provides insights into its potential roles in the abalone gut)

Dear Dr. Zongze Shao:

Link Not Available

Sincerely,

Konstantinos Kormas

Journals Department
Reviewer comments:

Reviewer #2 (Comments for the Author):

The manuscript reported a new strain of *Psychrilyobacter haliotis* B1, which has potential characteristics for probiotic candidates. The findings in the manuscript are very interesting and have a lot of novelties. However, there are several points in the "Materials and Methods" section which need to be addressed in order to be published, which are:

- Line 437, scientific name/species of brown seaweed and red algae should be added
- Line-438-439, the Size of abalone used in the present samples should be stated more specifically such as average size {plus minus} Sdev, and how the sampling process (anesthetize? in situ or laboratory ??...etc)

- Line 440, specific dilution of stomach content should be stated
- Protocol for the sampling process should be added.
- General suggestion

most of the methods to reveal various potential characteristics of *Psychrilyobacter haliotis* B1 were based on molecular approaches such as gene detections for fermenting monosaccharides and disaccharides, proteins, peptides, and amino acids, etc. However, there is no guarantee that these genes are active or being expressed, which is frequently reported in several previous studies. Therefore, additional studies using culture-dependent approaches should be conducted to confirm all these probiotic properties.

Reviewer #3 (Comments for the Author):

Comments

The authors had isolated the first host-associated *Psychrilyobacter* species from abalone gut and try to uncover its functional potentials to the host through different mechanisms. The results will contribute to the further understanding of *Psychrilyobacter* and complement the species of the species. It has important research significance and reference value. But the current results do not really delve into the interaction mechanism with the host. The paper needs to be revised before it can be considered for publication. The comments are as follows,

1. English needs further improvement, e.g "It showed a strong preference for the guts of marine invertebrates, especially abalone, which generally persisted with high relative abundances." The meaning of the expression is not clear. And "we combined of high-throughput sequencing, isolating and genomic analyses to uncover its potential role in host abalone." These sentences need to be revised.
2. It is suggested that the title be modified to make the topic more specific.
3. Line41-42, ".....acting as a potential probiotic gut anaerobe dominating in diverse marine invertebrates." Almost all the analyses in this study focused on the analysis of strain genes, and there was no manifestation or description of the influence of abalone phenotype. Moreover, the source of the strain was isolated from the similar size of abalone individuals. Why is the strain considered to be a probiotic?
- 4.Line182-183, Is there any further confirmation on the speculation of new species?
5. Line300-301 "In contrast, no PL is encoded in the genomes of the other two type strains (Table S9)." What species are the other two strains? It is best to indicate the bacterial species in parentheses here.
6. Line435-436, Why are the samples of abalone different? Is the homogenate is single or multi-individuals mixing? It suggested in the results (line103-108) that there were obvious individual differences. Why is the proportion of *Psychrilyobacter* in the individual of the same size so different?
7. Line502-503, what's "Prodigal" and "barnnap" mean?
8. Fig.1 and line 847, The text description of figure A and B ate too simple. It is suggested to specify the species of abalone on the X axis or description of numeric identification. What is the classification level in figure A, because most categories of latin names in figure A are uncertain. The title of the figure is "Distribution of *Psychrilyobacter* in different hosts and environments", but why was the figure A not showed *Psychrilyobacter*? What is the purpose of juxtaposing these two figures for the classification of intestinal flora of different abalones? And it's confusingthe while the same color in the two diagrams represents different bacteria. It should be clarified that the genus in figure B corresponds to the classification level in figure A.
9. Fig.2 The enrichment is carried out under anaerobic condition, why do a large proportion of aerobic bacteria such as *Vibrio* and *Shewanella* appear?
10. Fig.4 This photo is not clear, and the background is not clean enough, it is suggested a better photo be provided.

Staff Comments:

Preparing Revision Guidelines

Please return the manuscript within 60 days; if you cannot complete the modification within this time period, please contact me. If you do not wish to modify the manuscript and prefer to submit it to another journal, please notify me of your decision immediately so that the manuscript may be formally withdrawn from consideration by Microbiology Spectrum.

The manuscript reported a new strain of *Psychrilyobacter haliotis* B1, which has potential characteristics for probiotic candidates. The findings in the manuscript are very interesting and have a lot of novelties. However, there are several points in the “Materials and Methods” section which need to be addressed in order to be published, which are:

- Line 437, scientific name/species of brown seaweed and red algae should be added
- Line-438-439, the Size of abalone used in the present samples should be stated more specifically such as average size \pm Sdev, and how the sampling process (anesthetize? in situ or laboratory ??...etc)
- Line 440, specific dilution of stomach content should be stated
- Protocol for the sampling process should be added.
- General suggestion

most of the methods to reveal various potential characteristics of *Psychrilyobacter haliotis* B1 were based on molecular approaches such as gene detections for fermenting monosaccharides and disaccharides, proteins, peptides, and amino acids, etc. However, there is no guarantee that these genes are active or being expressed, which is frequently reported in several previous studies. Therefore, additional studies using culture-dependent approaches should be conducted to confirm all these probiotic properties.

Comments

The authors had isolated the first host-associated *Psychrilyobacter* species from abalone gut and try to uncover its functional potentials to the host through different mechanisms. The results will contribute to the further understanding of *Psychrilyobacter* and complement the species of the species. It has important research significance and reference value. But the current results do not really delve into the interaction mechanism with the host. The paper needs to be revised before it can be considered for publication. The comments are as follows,

1. English needs further improvement, e.g “It showed a strong preference for the guts of marine invertebrates, especially abalone, which generally persisted with high relative abundances.” The meaning of the expression is not clear. And “we combined of high-throughput sequencing, isolating and genomic analyses to uncover its potential role in host abalone.” These sentences need to be revised.
2. It is suggested that the title be modified to make the topic more specific.
3. Line41-42, “.....acting as a potential probiotic gut anaerobe dominating in diverse marine invertebrates.” Almost all the analyses in this study focused on the analysis of strain genes, and there was no manifestation or description of the influence of abalone phenotype. Moreover, the source of the strain was isolated from the similar size of abalone individuals. Why is the strain considered to be a probiotic?
4. Line182-183, Is there any further confirmation on the speculation of new species?
5. Line300-301 “In contrast, no PL is encoded in the genomes of the other two type strains (Table S9).” What species are the other two strains? It is best to indicate the bacterial species in parentheses here.
6. Line435-436, Why are the samples of abalone different? Is the homogenate is single or multi-individuals mixing? It suggested in the results (line103-108) that there were obvious individual differences. Why is the proportion of *Psychrilyobacter* in the individual of the same size so different?
7. Line502-503, what’s “Prodigal” and “barnnap” mean?
8. Fig.1 and line 847, The text description of figure A and B ate too simple. It is suggested to specify the species of abalone on the X axis or description of numeric identification. What is

the classification level in figure A, because most categories of latin names in figure A are uncertain. The title of the figure is "Distribution of *Psychrilyobacter* in different hosts and environments", but why was the figure A not showed *Psychrilyobacter*? What is the purpose of juxtaposing these two figures for the classification of intestinal flora of different abalones? And it's confusing the while the same color in the two diagrams represents different bacteria. It should be clarified that the genus in figure B corresponds to the classification level in figure A.

9. Fig.2 The enrichment is carried out under anaerobic condition, why do a large proportion of aerobic bacteria such as *Vibrio* and *Shewanella* appear?
10. Fig.4 This photo is not clear, and the background is not clean enough, it is suggested a better photo be provided.

Dear Dr. Konstantinos Aristomenis Kormas,

Thank you very much for your wonderful and constructive comments and suggestions. We would like to submit our revised manuscript entitled “Genomic and metabolic insights into the first host-associated isolate of *Psychrilyobacter*” (Spectrum03990-22R1). We have revised the manuscript to address the comments of the reviewers. We appreciate the reviewers' comments.

We have detailed our alterations point-by-point based on the reviewers' comments. Please see the details in the following “Response to reviewers' comments”. Each change has been red-marked in the “Marked Up Manuscript File”. In addition, other little changes on the English language and the Reference format are also red-marked in the manuscript.

Yours sincerely,

Zongze Shao

Response to reviewers' comments (Reviewers' comments are *italic* and our responses are marked in **blue**):

Reviewer 2:

*The manuscript reported a new strain of *Psychrilyobacter haliotis* B1, which has potential characteristics for probiotic candidates. The findings in the manuscript are very interesting and have a lot of novelties. However, there are several points in the “Materials and Methods” section which need to be addressed in order to be published, which are:*

- 1. Line 437, scientific name/species of brown seaweed and red algae should be added*

Response:

Thank you for your good suggestion. The scientific name of brown seaweed and red algae are *Laminaria japonica* and *Gracilaria lemaneiformis*, respectively. We have added this in the revised manuscript (Line 447).

2. *Line-438-439, the Size of abalone used in the present samples should be stated more specifically such as average size \pm Sdev, and how the sampling process (anesthetize? In situ or laboratory ??...etc)*

Response:

Thank you for the suggestions. We have added “The sizes of the abalone shells are approximately 5.5 ± 0.5 cm” in Line 448. To exclude the potential influence on the gut microbiota, we didn’t anesthetize to the abalone. The abalones were transported to the laboratory on ice immediately, and then the following operations were carried out in laboratory as soon as possible. More details on the sampling process are supplemented in Line 445-456 (red-marked).

3. *Line 440, specific dilution of stomach content should be stated*

Response:

Thank you for the advice. After grinding, we added 900 μ L of sterile artificial seawater to dilute the content. This part was changed in Line 450-451.

4. *Protocol for the sampling process should be added.*

Response:

As mentioned above, we have added the sampling process in Line 444-446. Thanks for your good suggestion.

General suggestion

*Most of the methods to reveal various potential characteristics of *Psychrilyobacter haliotis* B1 were based on molecular approaches such as gene detections for fermenting monosaccharides and disaccharides, proteins, peptides, and amino acids, etc. However, there is no guarantee that these genes are active or being expressed, which is frequently reported in several previous studies. Therefore, additional studies using culture dependent approaches should be conducted to confirm all these probiotic properties.*

Response:

We agree with the reviewer’s suggestion. Genomic-based inference cannot indicate the actual activity of the strain B1. Therefore, we added experiments to confirm the

fermentation products of different substrates by *P. haliotis* B1.

After the strain B1 cultured with glucose, mannitol, trehalose or tryptone as the only carbon source, the HPLC tests were performed to detect the organic acids using the formic acid, acetic acid, propionic acid, butyric acid, and lactic acid as the standards. As a result, the fermentation products of tryptone by strain B1 contained the short-chain fatty acids (e.g., acetic acid, propionic acid and butyric acid), providing these probiotics to the host. Thus, the culture-dependent results preliminarily confirmed the metabolic activity of strain B1 during fermenting tryptone (proteins, peptides or amino acids). In contrast, no such short-chain fatty acids were detected in the fermentation products in case of carbohydrates. Moreover, the strain B1 grew slowly with carbohydrates.

We have added the result in Line 350-358 and the methods in Line 536-544 of the revised manuscript, and the detailed methods and results are shown in the supplementary files. Thank you for the constructive suggestion again.

Reviewer 3

The authors had isolated the first host-associated Psychrilyobacter species from abalone gut and try to uncover its functional potentials to the host through different mechanisms. The results will contribute to the further understanding of Psychrilyobacter and complement the species of the species. It has important research significance and reference value. But the current results do not really delve into the interaction mechanism with the host. The paper needs to be revised before it can be considered for publication. The comments are as follows,

1. *English needs further improvement, e.g “It showed a strong preference for the guts of marine invertebrates, especially abalone, which generally persisted with high relative abundances.” The meaning of the expression is not clear.*

And “we combined of high-throughput sequencing, isolating and genomic analyses to uncover its potential role in host abalone.” These sentences need to be revised.

Response:

Thank you for your suggestion.

First, we have rephrased the unclear expression. We merged this sentence with the front related one and removed some redundancies. The new sentence is “The high-throughput sequencing and literature compiling results indicated that *Psychrilyobacter* is widely distributed in marine and terrestrial ecosystems with both host-associated and free-living lifestyles, but with a strong niche preference in the guts of marine invertebrates, especially abalone” in Line 25-28. And we have revised another sentence to “we combined cultivation-independent and cultivation-dependent methods to uncover the potential roles of *Psychrilyobacter* in the host abalone” in Line 23-25.

English expression was checked by a scholar efficient in English. Finally, we have polished our manuscript in a professional editing service company (American Journal Experts, AJE). The “editing certificate” screenshot is attached below.

We hope these efforts will improve the English.

2. *It is suggested that the title be modified to make the topic more specific.*

Response:

Thank you for the good suggestion. Considering our study mainly focused on the genomic analyses and metabolic features of the first host-associated pure culture of *Psychrilyobacter*, we decide to modify the title to “Genomic and metabolic insights into the first host-associated isolate of *Psychrilyobacter*”.

3. *Line41-42, “.....acting as a potential probiotic gut anaerobe dominating in diverse marine invertebrates.” Almost all the analyses in this study focused on the analysis of strain genes, and there was no manifestation or description of the influence of abalone phenotype. Moreover, the source of the strain was isolated from the similar size of abalone individuals. Why is the strain considered to be a probiotic?*

Response:

Thank you for your comments. Yes, we agree with the reviewer’s opinion here. First, our study showed genomic and metabolic evidence from the bacterium, like producing of SCFAs and synthesis of vitamins, which have been proved to be beneficial to hosts in previous related studies (Please see the discussion in Line

335-349 on the SCFAs and Line 375-391 on the vitamins). Notably, we have supplemented additional experiment (HPLC) to test the ability of producing SCFAs by the strain under carbohydrates or tryptone conditions. As a result, diverse and abundant SCFAs could be produced by the strain when the tryptone was fermented (Please see that in Line 352-355). The result further supports its beneficial potential through SCFAs in the host gut.

Second, evidence from other previous studies also have shown that *Psychrilyobacter* is widely and abundantly distributed in healthy abalone gut (Wang et al., 2020), and it was more prevalent in abalones with larger size (Choi et al., 2021). We have discussed this in our manuscript in Line 415-418. In the current study, we paid more attention to the genomic and metabolic features of the pure culture, and studies on the relationship between abalone phenotype and *Psychrilyobacter* will be carried out in the future.

Finally, we have pointed out and discussed that the strain B1 cannot be defined as a probiotic in Line 420-423.

Considering the above reasons, we have removed the word ‘probiotic’, and rephrased the sentence to “....., genomic and metabolic evidence showed some beneficial characteristics of the dominant gut anaerobe to the host” in Line 41-42.

4. *Line182-183, Is there any further confirmation on the speculation of new species?*

Response:

Thank you for your question. The whole-genome-based ANI (average nucleotide identity) and DDH (DNA–DNA hybridization) are the two golden rules in the delineation of novel bacterial species (Jain et al., 2018; Wayne et al., 1987), as shown in Line 186-189. Furthermore, we also compared the phenotypic characteristics of strain B1 with the other two cultured Type strains of *Psychrilyobacter*, *P. atlanticus* HAW-EB21^T and *P. piezotolerans* SD5^T in Line 197-218. All the evidence has shown that *P. haliotis* B1 is a new species.

5. *Line300-301 “In contrast, no PL is encoded in the genomes of the other two type strains (Table S9).” What species are the other two strains? It is best to indicate the bacterial species in parentheses here.*

Response:

Thank you for the good suggestion. The two type strains are *P. atlanticus* HAW-EB21^T and *P. piezotolerans* SD5^T, we have added in Line 303-304.

6. Line435-436, Why are the samples of abalone different? Is the homogenate is single or multi-individuals mixing? It suggested in the results (line103-108) that there were obvious individual differences. Why is the proportion of *Psychrilyobacter* in the individual of the same size so different?

Response:

Thank you for the questions.

For the first question: We sampled different abalone species mainly to increase the probability of obtaining the pure culture of the *Psychrilyobacter*. We choose two kinds of Chinese local abalone to test the *Psychrilyobacter* abundance and decide to use which kind of abalone for the further enrichment and isolation.

For the second question: For the microbial diversity analysis, the homogenate is single individual, in order to quantify the percentage of *Psychrilyobacter* in each abalone gut. For the enrichment, we mixed the two individuals of *H. discus hannai* homogenate together, we can not only increase the chance to get the strain but also can decrease the workload. We have rephrased the sentence to “After adding 900 μ L of sterile artificial seawater, the intestine of each abalone individual was thoroughly grinded to homogenate. For the bacterial diversity analysis, the individual homogenate of *H. discus hannai* and *H. diversicolor* was preserved at -80°C until the DNA extraction. For the bacterial enrichment, the homogenate of two mixed individuals of *H. discus hannai* was directly used without freezing as described below.” in Line 450-455.

For the Third question: Through the Fig. 1C, both our samples and other previous reports showed that the proportion of *Psychrilyobacter* varied from about 10% to 90%, indicating that the individual differences are common in the abalone guts. However, the reason remains unclear to date. We guess that this may be a result of multi-factors, such as the food, healthy condition, species and other abiotic and/or biotic factors. This is really an interesting question, and further

investigations involved in the bacterial community and multi-factors maybe give the answer in the future.

7. Line 502-503, what's "Prodigal" and "barrnap" mean?

Response:

Thank you for the questions. Both "Prodigal" and "barrnap" refer to two computing programs we used, Prodigal for "Prokaryotic Dynamic Programming Gene-finding Algorithm" and barrnap for "Basic Rapid Ribosomal RNA Predictor". We have added their full names in Line 515-517.

8. Fig.1 and line 847, The text description of figure A and B are too simple. It is suggested to specify the species of abalone on the X axis or description of numeric identification. What is the classification level in figure A, because most categories of latin names in figure A are uncertain. The title of the figure is "Distribution of *Psychrilyobacter* in different hosts and environments", but why was the figure A not showed *Psychrilyobacter*? What is the purpose of juxtaposing these two figures for the classification of intestinal flora of different abalones? And it's confusing the while the same color in the two diagrams represents different bacteria. It should be clarified that the genus in figure B corresponds to the classification level in figure A.

Response:

According to your good suggestions, we have added the description of the species of abalone "The abalone individual number 1 and 2 are *H. discus hannai*, and the individual number 3 to 5 are *H. diversicolor*" in the figure legend (Line 874-875). The classification in figure A is at the phylum level, we have revised this in the figure legend in Line 872-873.

We are sorry for the ambiguities here. We have changed the title to "Bacterial community compositions in the sampled abalones' guts and the distribution of *Psychrilyobacter* in different hosts and environments". Figures A and B represent the bacterial communities at the phylum and genus level, respectively. We have revised the legend to "(A) The relative abundance of gut bacteria in the sampled abalone at the phylum level. (B) The relative abundance of gut bacteria in the

sampled abalone at the genus level". Meanwhile, we have clearly annotated the meaning of each color at the right of Figures A and B, respectively.

Thank you for the good suggestion, and we hope the changes making the figure clearly.

9. *Fig.2 The enrichment is carried out under anaerobic condition, why do a large proportion of aerobic bacteria such as Vibrio and Shewanella appear?*

Response:

Thank you for the question. Actually, *Vibrio* and *Shewanella* are both facultative anaerobes instead of obligate anaerobes; they are able to grow either with or without free oxygen. For example, Sawabe and colleagues have isolated two *Vibrio* species from the gut of Japanese abalones (Sawabe et al. 2004), and *Shewanella* species was also successfully isolated from the gut of the abalone, *Haliotis discus hannai* (Kim et al., 2007). Therefore, *Vibrio* and *Shewanella* appeared in the enrichment of abalone gut microbiota should be a normal phenomenon here.

10. *Fig.4 This photo is not clear, and the background is not clean enough, it is suggested a better photo be provided.*

Response:

According to the good suggestion, we have re-cultured the strain B1 in anaerobic condition and photographed its morphology through the transmission electron microscopy (TEM) after negative staining again. We have replaced the unclear photo by the new one, and it is attached below.

x8.0k Zoom-1 HC-1 80.0kV 2023/05/31 13:34
Hitachi TEM system. 1.0um

References

- Choi M J, Oh Y D, Kim Y R, et al. Intestinal microbial diversity is higher in Pacific abalone (*Haliotis discus hannai*) with slower growth rates. *Aquaculture*, 2021, 537: 736500.
- Wang X, Tang B, Luo X, et al. Effects of temperature, diet and genotype-induced variations on the gut microbiota of abalone. *Aquaculture*, 2020, 524: 735269.
- Wayne L G, Brenner D J, Colwell R R, et al. Report of the ad hoc committee on reconciliation of approaches to bacterial systematics. *International Journal of Systematic and Evolutionary Microbiology*, 1987, 37(4): 463-464.
- Jain C, Rodriguez-R L M, Phillippy A M, et al. High throughput ANI analysis of 90K prokaryotic genomes reveals clear species boundaries. *Nature communications*, 2018, 9(1): 5114. Sawabe T, Hayashi K, Moriwaki J, et al. *Vibrio neonatus* sp. nov. and *Vibrio ezurae* sp. nov. isolated from the gut of Japanese abalones. *Systematic and applied microbiology*, 2004, 27(5): 527-534.
- Kim D, Baik K S, Kim M S, et al. *Shewanella haliotis* sp. nov., isolated from the gut microflora of abalone, *Haliotis discus hannai*. *International journal of systematic and evolutionary microbiology*, 2007, 57(12): 2926-2931.

August 11, 2023

Prof. Zongze Shao
Third Institute of Oceanography Ministry of Natural Resources
Key Laboratory of Marine Biogenetic Resources
Daxue Road 178#
Xiamen, Fujian 361005
China

Re: Spectrum03990-22R1 (Genomic and metabolic insights into the first host-associated isolate of *Psychrilyobacter*)

Dear Prof. Zongze Shao:

Your manuscript has been accepted, and I am forwarding it to the ASM Journals Department for publication. You will be notified when your proofs are ready to be viewed.

Sincerely,

Konstantinos Kormas
Editor, Microbiology Spectrum
